# Exploring the Three-Dimensional Frontier: Advancements in MSC Spheroids and Their Implications for Breast Cancer and Personalized Regenerative Therapies

**DOI:** 10.3390/biomedicines12010052

**Published:** 2023-12-25

**Authors:** Veronika Smolinska, Stefan Harsanyi, Martin Bohac, Lubos Danisovic

**Affiliations:** 1Institute of Medical Biology, Genetics and Clinical Genetics, Faculty of Medicine, Comenius University in Bratislava, Sasinkova 4, 811 08 Bratislava, Slovakia; smolinska7@uniba.sk (V.S.); martin.bohac@fmed.uniba.sk (M.B.); lubos.danisovic@fmed.uniba.sk (L.D.); 2Regenmed Ltd., Medena 29, 811 02 Bratislava, Slovakia

**Keywords:** stem cells, MSCs, 3D modeling, spheroids, breast cancer, personalized therapy

## Abstract

To more accurately replicate the in vivo three-dimensional (3D) mesenchymal stem cell (MSC) niche and enhance cellular phenotypes for superior in vivo treatments, MSC functionalization through in vitro 3D culture approaches has gained attention. The organization of MSCs in 3D spheroids results in altered cell shape, cytoskeleton rearrangement, and polarization. Investigations have revealed that the survival and secretory capability of MSCs are positively impacted by moderate hypoxia within the inner zones of MSC spheroids. The spheroid hypoxic microenvironment enhances the production of angiogenic and anti-apoptotic molecules, including HGF, VEGF, and FGF-2. Furthermore, it upregulates the expression of hypoxia-adaptive molecules such as CXCL12 and HIF-1, inhibiting MSC death. The current review focuses on the latest developments in fundamental and translational research concerning three-dimensional MSC systems. This emphasis extends to the primary benefits and potential applications of MSC spheroids, particularly in the context of breast cancer and customized regenerative therapies.

## 1. Introduction

Mesenchymal stem/stromal cells (MSCs), initially discovered in bone marrow by Friedenstein et al. (1974), have since been identified in various connective tissues and bodily fluids, including adipose tissue, synovium, synovial fluid, and the umbilical cord [1,2,3,4]. Their significance in cell therapy lies in the straightforward isolation process, considerable proliferative capacity in vitro while maintaining stemness, and notable paracrine immunomodulatory and trophic actions in vivo. These qualities contribute to the perceived advantage of MSCs in therapeutic applications [5]. Over the past 30 years, the safety profile of MSCs has been well demonstrated in clinical trials for treating diverse conditions, with some trials indicating positive outcomes. Beyond these clinical applications, efforts have been directed toward overcoming limitations in traditional in vitro/ex vivo approaches. Specifically, researchers aim to create programs that generate specific characteristics, traits, and functionalities in developing cells to extend their potential applications [6]. To more accurately replicate the in vivo three-dimensional (3D) MSC niche and enhance cellular phenotypes for superior in vivo treatments, MSC functionalization through in vitro 3D culture approaches has gained attention. The organization of MSCs in 3D spheroids results in altered cell shape, cytoskeleton rearrangement, and polarization. The process involves cell aggregation and subsequent multicellular spheroid development, influenced by factors like membrane-bound integrin interactions and cadherin gene expression levels (Figure 1) [6].

Novel investigations have revealed that the survival and secretory capability of MSCs are positively impacted by moderate hypoxia within the inner zones of MSC spheroids. The spheroid hypoxic microenvironment enhances the production of angiogenic and anti-apoptotic molecules, including HGF, VEGF, and FGF-2. Furthermore, it upregulates the expression of hypoxia-adaptive molecules such as CXCL12 and HIF-1, inhibiting MSC death [7]. Spheroid size emerges as a significant factor influencing nutrient and oxygen supply, as well as mechanical stresses arising from cell-to-cell interactions. This, in turn, modifies gene expression and underscores the importance of carefully controlling the 3D culture environment for optimal outcomes, as well as choosing the most suitable spheroid generation technique (Figure 2) [8]. The utilization of 3D MSC models has provided valuable insights into the fundamental characteristics of MSCs and has paved the way for the development of diverse treatment approaches. These approaches influence the multilineage differentiation capability and secretory activity of MSCs for therapeutic benefits (Table 1).

Beyond the realm of MSCs, three-dimensional cell culture models have proven revolutionary, particularly in cancer drug research. These models, reflecting the three-dimensional nature of tissues and tumors, offer a more precise understanding of illnesses compared to traditional two-dimensional cell cultures and animal models. Recent research has underscored the superiority of 3D models in simulating physiological environments and in vivo situations [9]. Organoids, i.e., three-dimensional structures derived from stem cells, have gained prominence in this culture method. Organoids replicate several biological interactions observed in vivo and serve as miniature versions of organs, maintaining their original shape and architecture. This technology is now extensively employed for disease modeling and drug screening [10,11].

The utilization of 3D culture strategies has proven beneficial in improving the survival and immune-modulatory effects of MSCs. In a 3D culture environment, cells produce pro-inflammatory cytokines in larger quantities, which stimulate MSCs to release various soluble factors. These factors play a crucial role in controlling immune cells, reducing inflammation, and aiding in tissue repair [12]. The growing interest in 3D MSC platforms arises from the desire not only to control cellular interactions in vitro, but also to consistently produce stable cartilage constructs resembling hyaline. Overcoming challenges in MSC-based cartilage regeneration involves the development of enhanced platforms that address translational hurdles, ultimately improving therapeutic outcomes [13]. Cell sheet technology shows promise in the repair and replacement of hyaline cartilage using diverse cell sources and preparation techniques. By cultivating chondrogenically differentiated MSC sheets in vitro, these sheets can be directly transplanted in vivo, potentially expediting the regeneration process [14]. In adherent 2D conditions, where cells are initially sown and developed, a sudden temperature-mediated detachment causes existing cytoskeletal filaments and the retained extracellular matrix to spontaneously constrict. This results in the formation of a 3D, multi-nucleus, thick, scaffold-free cell sheet structure. The post-detachment cell sheet contraction is vital for creating an environment conducive to spontaneous engraftment in tissue locations and the rapid initiation of direct cell–cell communication [14,15].

In the realm of regenerative medicine, the use of biomaterials is crucial. Biomaterials play a key role in tissue engineering, providing instructive signals to guide cell function and promoting faster drug delivery. Incorporating biomaterials can reduce the time spent in ex vivo culture, enabling the quicker delivery of medicines while using fewer resources [16,17]. The evolving landscape of 3D bioprinting, an additive manufacturing method, has emerged as a potential solution for creating transplant-ready tissues and organs. By layering live cells and growth factors in a precise manner, 3D bioprinting offers a means to design and produce complex, functional structures. The process involves designing the 3D structure using computer modeling; printing using bio-ink; and evaluating the physical, mechanical, and biological properties before transplantation [18].

Choosing the appropriate bio-ink and cell type is critical in 3D bioprinting. The bio-ink material must possess the correct signaling proteins, mechanical and structural qualities, and growth and adhesion factors. The selection of cells includes choosing those with the ability to proliferate, differentiate within the printed scaffold, communicate with signaling molecules, survive the printing process, and persist afterward. Recent developments in 3D bioprinting have shown promise in creating artificial skin, aiding burn victims, and restoring damaged cartilage [19]. The exploration of changeable biomaterials suggests a viable method for directing and maintaining spheroid differentiation, simulating tissue-specific developmental processes. While this avenue is still being explored, the potential impact on endogenous restorative systems is intriguing [20]. These findings underscore the multifaceted applications and the ever-expanding potential of 3D cell culture and tissue engineering in the realms of regenerative medicine and personalized therapeutics.

The current review focuses on the latest developments in fundamental and translational research concerning three-dimensional MSC systems. This emphasis extends to the primary benefits and potential applications of MSC spheroids, particularly in the context of breast cancer and customized regenerative therapies.

## 2. Three-Dimensional Models in Breast Cancer

The most common kind of cancer among women is acknowledged to be breast cancer, which has a considerable influence on both longevity and quality of life. Traditional treatments, namely, surgery, radiation, and chemotherapy, help to manage the cancer, but do not provide a long-term cure [21]. Numerous signals coming from the environment around cancer cells have a significant impact on how the tumor develops. It has been proposed that bone marrow may operate as a storage location for dormant tumor cells that recirculate and infect other organs when conditions are suitable [22]. However, studies have shown both tumor-promoting and tumor-suppressive responses, making the impact of MSCs on tumor growth inconsistent [23]. Collectively, it is clear that understanding the mechanisms behind MSC-mediated modification of tumor cell activity is essential for both developing novel, precise therapies and ensuring the safe application of MSCs in the clinic. To better understand MSC biology and improve MSC-based therapeutics, there has been a rising interest in employing 3D nonadherent culture platforms, which are often employed as tumor models and in drug development [24,25,26,27]. A unique platform for high-throughput analysis of molecular alterations during the start, development, and metastasis of breast cancer may be provided by the use of 3D cell culture models, which have improved our understanding of how breast cancer progresses and allowed us to test new therapies. However, new models are required which yield reproducible quantitative data that better mimic the mammary gland architecture. Early studies have shown that mammary morphogenesis, mammary gland branching, and differentiation of normal and cancerous mammary phenotypes are assayable domains utilizing a laminin-rich matrix, also known as Matrigel or a collagen matrix [28,29,30]. The foundation for current 3D culture practices for breast models was set by follow-up investigations [31]. Importantly, when MSCs’ secretomes or conditioned media (CM) are delivered to tumor cells, a general anti-tumor impact is frequently seen, tipping the scales in favor of the creation of innovative cell-free-based treatments [32,33].

### 2.1. MSCs as a Tool towards Novel Anti-Cancer Therapeutic Targets

Non-adherent cultures have been utilized successfully to explore certain cell engulfment programs, often known as cell cannibalism, according to a paper by Bartosh et al. Generally speaking, cell cannibalism—also known as xenocannibalism—describes a process in which a cell encloses and ultimately destroys one or more target cells that are either nearby and of the same (homotypic) kind or of a distinct (heterotypic) kind [34]. Cell cannibalism, a live-cell feeding activity, is regarded to be different from the traditional phagocytosis that macrophages utilize to destroy apoptotic cells from a molecular standpoint [35]. Although one result of entosis, akin to cell cannibalism, is the death of the internalized cell, it is also thought to be separate from the live-cell engulfment processes of entosis and emperipolesis, which require active invasion/penetration of one cell into the cytoplasm of another [36,37]. The researchers focused on the interactions between bone marrow-derived MSCs and MDA-MB-231 (MDA) breast cancer cells (BCCs) in this investigation using hanging drop cultures. The results showed that BCCs under stress could consume or “cannibalize” MSCs in these 3D cocultures, a process that mirrored, from a morphological perspective, the rare but well-documented clinical phenomenon of cancer cell cannibalism and that boosted inflammatory response and cell sustainability while stalling tumor formation. The study demonstrated that, after consuming bone marrow-derived MSCs, the BCCs had a robust ability to endure in environments with restricted food supply, as predicted, but their capacity to develop tumors in mice was reduced. Together, the findings showed that following inoculation, the BCCs developed a phenotype that was typical of dormant cancer cells or, at the very least, promoted tumor dormancy [34].

Because there was no evidence of cell cannibalism in 2D adherent cultures or in vivo following coinjections of MDA cells and MSCs, the 3D tumor niche model was crucial for enhancing cell feeding behaviors and determining the effects of cannibalism. Microarray experiments were used to further evaluate the MDA phenotype, and it was shown that cannibalism of MSCs caused an up-regulation of several cytokines and chemokines. Inflammatory mediators like IL-1, IL-1, IL-6, IL-8, CXCL1, CXCL2, GCSF, and PAI-1 (SERPINE1), all of which were up-regulated after MSC cannibalism, are byproducts of senescent cells and are important components of the senescence-associated secretory phenotype [38,39]. Even though the role of inflammation in tumor progression has been controversial, the results were intriguing. Both phenomena (i.e., cell cannibalism and dormancy) involve cell survival strategies, are linked to growth arrest, are most frequently seen in highly aggressive cancers, and are represented by minor/residual cells. The results suggest that cannibalism of MSCs within the tumor niche represents a unique mechanism supporting cancer dormancy. The team has discovered a particular and distinctive cancer phenotype linked to BCC-MSC cross-talk that may provide new opportunities for medical treatment. The result also implies that a realistic 3D coculture model can be a beneficial tool for understanding and utilizing MSCs’ antitumor capabilities and cell cannibalism in the future [34]. Future research is required in order to pinpoint the specific processes behind the MSC cannibalism of cancer cells.

The development, growth, and invasiveness of cancer depend heavily on interactions between tumor cells and their microenvironment. Mesenchymal stem cells in particular are attracted to areas of growing malignancies, encouraging the development of metastases. Although the migration and integration of MSCs in the tumor microenvironment (TME) are well documented, the role of MSCs is still unclear after they reach the tumor. According to data, breast cancer cells that interact with MSCs proliferate and metabolically function more actively, in part because of MSC-derived microvesicles that are released into the TME [40,41]. Maffey et al. explored the link between MSCs and tumor cells as well as the possibility of influencing P2X-mediated intercellular communication to alter this association [42]. A tonic activation of P2X7, which is associated with an anti-apoptotic and growth-promoting action, is ensured by micromolar amounts of ATP at the extracellular site. However, a rising body of research contends that, rather than having a cytotoxic impact, the tonic activation of the P2X7 receptor has trophic, growth-promoting properties [41]. Numerous studies and many cancer types have shown a connection between cancer and purinergic receptors. When activated by high (millimolar) amounts of ATP, the P2X7 receptor is recognized as being a key factor in cellular death by apoptosis or necrosis [40]. Researchers were able to decrease the cancerogenic and metastatic potential of breast cancer cells cocultured with human adipose-derived MSCs in both 2D and 3D in vitro models by suppressing P2X-mediated purinergic signaling (measured via microfluidic chips, followed by analysis via mammosphere formation assay). The collected data showed, for the first time, that MSCs have a trophic influence on breast cancer cell proliferation and that this effect is mediated by ionotropic purinergic signaling, hinting that purinergic signaling system suppression may be a viable therapeutic target [42].

It is unclear exactly how MSCs cause cancer cells to invade. Using a 3D coculture model, McAndrews and his team examined how MSCs influence the migration of invasive breast cancer cells. Breast cancer cells’ elongation, directional migration, and traction creation are all increased in coculture with MSCs. Transforming growth factor (TGF-β) and the migratory proteins Rho-associated kinase, focal adhesion kinase, matrix metalloproteinases, and MSC-induced directional migration all directly connect with traction production. Similar migration results were measured after treatment with recombinant TGF- β1 or MSC-conditioned medium. Together, these studies demonstrate that MSCs release TGF-β, which causes invasive breast cancer cells to migrate in a force-dependent manner. These pathways might serve as potential targets for preventing the invasion and subsequent spread of cancer cells [43].

It was previously demonstrated that MCF7, the cell type employed by McAndrews et al., was mainly non-motile in collagen gels, most likely because it had an epithelial character [44]. Epithelial-to-mesenchymal transition (EMT), which results in a more motile phenotype, can be seen in breast cancer cells treated with MSC conditioned media or TGF-β; however, this transformation takes place after 3–7 days of exposure to these stimuli [45,46]. Although MCF7 cells exhibit certain EMT markers, heterogenous expression of E-cadherin and vimentin was detected as well. This indicated that these cells had not yet fully made the transition to a mesenchymal phenotype. The chromatin structure of MCF7 cells does not permit complete EMT, according to other research [47]. Together, these data show that the epithelial phenotype of MCF7 cells prevents MSCs from inducing the migration of MCF7 cells on the time scale examined in motility assays. Longer coculture trials may lead to EMT and MCF7 cell migration. According to the study’s results, MSCs release TGF-β, which activates ROCK, FAK, and MMPs to cause the directional migration of MDA cells. Together, these studies shed light on how MSCs interact with aggressive breast cancer cells in the tumor microenvironment and suggest possible treatment targets to prevent metastasis and invasion [43].

In their most recent study, Melzer et al. showed how coculturing MSCs with malignant breast cancer cells causes changes in the synthesis of urokinase plasminogen activator (uPA), plasminogen activator inhibitor type-1 (PAI-1), and TGF-β, which may encourage ongoing tumor development. The results show that long-term coculturing of human MSC with breast cancer cell line MDA-MB-231 affects the expression of uPA, PAI-1, and TGF-β1, three factors that promote the spread of cancer cells and boost the production of metastasis. Three-dimensional tumor spheroids were produced in vitro by long-term direct coculture of benign human mesenchymal stroma/stem-like cells MSC544^GFP^ and human MDA-MB-231^cherry^ breast cancer cells, expressing mCherry and eGFP fluorescence proteins, respectively [48]. When measured in comparison to monocultures of normal human MSCs, the breast tumor marker uPA was shown to be more abundant in the monoculture of MDA-MB-231 cells. Additionally, uPA levels in 3D tumor spheroids persisted at levels 9.4 times higher than the human MSC monocultures. The expression of PAI-1 was lower by 2.6-fold in the breast cancer cells compared to the MSC monoculture. Comparing the MDA-MB-231^cherry^/MSC544^GFP^ 3D coculture spheroids to the MSC populations, PAI-1 was reduced by 5.8-fold. The synthesis of TGF-β1, a key growth factor in regulating tumor development and metastasis formation, was higher in MDA-MB-231^cherry^/MSC544^GFP^ cocultures than in MSC544 monocultures after 24 h, and it increased significantly during the following 72 h. TGF-β1 is synthesized by MSCs instead of by breast cancer cells in MSC544 and MDA-MB-231 cells, and TGF-β1 also contributes to its production in these cells, according to quantitative PCR investigations. These results showed that the coculturing of breast cancer cells with MSCs could have synergistic effects on the expression/secretion of uPA, PAI-1, and TGF-β [48].

One of the most crucial areas of study for the treatment of illnesses, including cancer, infections, and autoimmune disorders, is drug discovery. The 2D cell culture system has so far been used to assess the therapeutic effects or the adverse effects of medications. However, the circumstances of 2D cell culture are very unlike those of an in vivo environment. The in vitro and in vivo drug effects frequently differ greatly because of this discrepancy, which leads to failed drug development [49]. Three-dimensional culture models have been studied lately to imitate in vivo circumstances. Hydrogel microspheres of gelatin (GM), a biodegradable natural polymer, have been investigated for incorporation into cell aggregates due to their increased functionalities and longer culture as one attempt to address the issue. This is because GM matrices’ aqueous phase allows nutrients as well as oxygen to enter into the aggregates [50,51]. The 3D cell aggregates integrating GM would constitute a potential cell culture system for drug development based on this characteristic. The goal of Nii et al.’s study was to explore polarized MSC groups—MSC1 (cancer diminishing type) and MSC2 (cancer supportive type). This research group created a coculture system for cancer cells and 3D MSC2 aggregates containing GM and assessed its effectiveness in cancer cell invasion in vitro. Three-dimensional MSC2 aggregates, including GM, were first created. Based on a comparison with other control groups, the MSC2 function was evaluated using the secretion of a chemokine (C-C motif) ligand (CCL) 5. The proliferation rate of cancer cells was assessed in vitro after the coculturing of cancer cells and 3D MSC2 aggregates, including GM. The team looked at matrix metalloproteinase (MMP)-2 production as a marker of tumor invasion promotion.

It has been shown that mesenchymal stem cells are also, in addition to macrophages, polarized to MSC1 or MSC2. While MSC2 is engaged in the proliferation, invasion, and metastasis of cancer cells, MSC1 can reduce the number of cancer cells or have an anti-cancer impact. MSC2 is, therefore, a key component cell in the cancer environment that helps cancer cells to survive or perform better. The release of chemokine (C-C motif) ligand (CCL)5 or interleukin (IL)-1RA has been noted as an MSC2 cancer-related function [52,53,54]. Based on the interaction between cancer cells and MSC, particularly the immunosuppressive MSC phenotype (MSC2), this study developed a drug-screening model [53]. First, a gelatin hydrogel microsphere 50 mm long was included among MSC2 aggregates. The aggregates were around 800 mm in size and almost spherical. As MSC2 began to work, the amount of chemokine (C-C motif) ligand 5 that was secreted increased. By secreting matrix metalloproteinases, the MSC2 aggregates, including GM, enhanced the capacity of cancer cells to invade. The ability to assess the in vitro invasion behavior of cancer cells using a 3D culture system of MSC2 containing GM seems encouraging. To assess the in vitro invasion behavior of cancer cells, a 3D culture system of MSC2 integrating GM and cancer cells appears promising [53].

### 2.2. MSC Secretome as a Co-Treatment with Traditional Therapy

Doxorubicin (Dox), which has historically been used as a chemotherapy drug to treat breast cancer, has side effects that are still of major clinical concern. The use of MSCs’ secretomes has been studied concerning its active function in immunomodulation and regeneration processes, highlighting its tremendous potential in many pathological disorders [55,56]. Its effects in a cancer environment have not yet been identified. The most recent investigation, published by Serras et al., sought to comprehend how therapeutically relevant dosages of Dox in conjunction with the secretomes of MSCs affect both tumor and non-tumor cells. Novel methods are required in order to clarify the use of CM from 2D and 3D MSC cultures in both cancer and normal-type cells [57]. In this regard, the current study sought to further clarify the putative underlying mechanisms through a thorough mechanistic proteomic analysis, aiming to assess the effects of the secretomes of MSCs in human malignant breast cells and non-tumor cells (i.e., normal breast epithelial cells and cardiomyocytes) upon co-treatment with Dox. All in all, the findings demonstrated that tumor and non-tumor cells responded differently to the secretomes of MSCs. The purpose of this study was to investigate the effects of the MSC-derived secretomes of 3D (CM3D) or 2D (CM2D) cultures in differentiated AC16 cardiomyocytes, human malignant breast cells (MDA-MB-231), and non-tumor breast epithelial cells (MCF10A) coincubated with Dox. The MSC secretomes did not alter the Dox’s cytotoxic, anti-migration, or anti-invasive actions when mixed with it. Additionally, concurrent treatment of MCF10A and AC16 cells resulted in a partial, but considerable, preservation of cell viability. The Dox-related cytotoxic impact persisted in MDA-MB-231 cells. Additionally, CM3D regularly outperformed CM2D, and numerous proteins were shown to be associated with these divergent outcomes [57]. This work envisioned a safer and more effective use of chemotherapeutic medicines while also advancing the development of innovative adjuvant anticancer therapy.

### 2.3. Innovation of the Scaffold-Free, 3D Breast Organoid Model

Three-dimensional cellular models are valuable for studying biological processes, while gel-embedded organoids exhibit a high degree of variability. The major benefits of the scaffold-free, 3D breast organoid model developed by Dr. Djomehri were presented in their latest study. High consistency and replication of the 3D model and the ability to measure cellular collagen I production without interference from exogenous collagen were reported with this model, as well as the ability to subject the organoid to a variety of microenvironmental and exogenous treatments at precisely timed intervals without worrying about matrix binding [58]. By using this approach, primary metaplastic mammary carcinomas from MMTV-Cre;Ccn6fl/fl mice were converted into organoids that preserved the high-grade spindle cell shape of the original tumors. The platform is intended to be used as a standardized 3D cellular model to investigate how breast carcinogenesis is influenced by the microenvironment and to investigate new treatments. In the current study, non-tumorigenic mammary MCF10A cells, MDA-MB-231 breast cancer cells, cocultures with MCF10A and MSC cells, and primary carcinomas from MMTV-cre;Ccn6fl/fl mice were cultured using a scaffold-free 384-well hanging drop system (Figure 2). According to the results, MCF10A cells were able to grow into organoids with cellular phenotypes that mirrored the structure of a normal human breast. In 3D suspension culture, it was discovered that laminin-rich components—rather than a laminin-rich matrix—along with a straightforward crowding agent and FBS are sufficient to support the formation of mammary acinar structures with high reproducibility. Employing scaffold-free culture MCF10A cells formed a sizable, self-organized organoid structure with the ability to display multiple lineage phenotypes [58]. The team was able to create neoplastic organoids that had repeatable, homogeneous sizes and shapes, making it easier to standardize them against other factors. The organoids suffered cellular-level phenotypic alterations and organoid-level morphologic changes that might be measured to stratify organoid responses when subjected to settings that resemble neoplastic development. High-throughput testing and assay standardization are made possible by the simplicity of the one droplet, one organoid method, the regular form and size of the produced organoids, and the drastic morphological alterations brought on by contact with neoplastic environments. The system is intended to be used as a standardized 3D cellular model to investigate how breast carcinogenesis is influenced by the microenvironment and to investigate new treatments. Additionally, the limited capacity of gel systems to express tissue-specific markers is a significant drawback. The information provided supports the hypothesis that differentiation could occur more easily in a 3D free-floating situation than was previously suggested. In order to support the desired outcome, scientists employing scaffold or suspension systems should take into account the pertinent biophysical and spatiotemporal aspects [58].

## 3. Conclusions

In conclusion, this comprehensive review underscores the evolving landscape of three-dimensional (3D) mesenchymal stem cell (MSC) systems and their profound implications for regenerative medicine, particularly in the context of breast cancer and personalized therapies. The integration of MSCs into 3D spheroids has proven instrumental in enhancing cellular phenotypes, with a focus on the advantageous effects of moderate hypoxia within the inner zones of these spheroids. The resultant microenvironment promotes the secretion of key angiogenic and anti-apoptotic molecules, contributing to the survival and therapeutic potential of MSCs.

The multifaceted applications of 3D cell culture extend beyond MSCs, encompassing revolutionary models in cancer drug research. Specifically, in the realm of breast cancer, 3D models provide a more precise understanding of the complex interactions between tumor cells and their microenvironment. This review delves into studies demonstrating the impact of MSCs on breast cancer progression, revealing intriguing phenomena such as cell cannibalism and dormancy. These findings not only enhance our comprehension of MSC-mediated modification of tumor cell activity, but also hold promise for innovative, cell-free-based treatments.

Moreover, the exploration of 3D culture strategies extends to biomaterials and 3D bioprinting, offering potential solutions for tissue engineering and organ transplantation. The use of biomaterials in tissue engineering guides cell function and facilitates drug delivery, while 3D bioprinting emerges as a groundbreaking method for creating complex, functional structures with applications in skin regeneration and cartilage repair.

This review further addresses the critical role of drug discovery in 3D cellular models, emphasizing the limitations of traditional 2D cell culture systems. Studies utilizing gelatin hydrogel microspheres and MSC aggregates showcase the potential of 3D models for drug development, presenting a more realistic representation of in vivo environments.

In summary, this review navigates through the intricate web of 3D MSC systems, providing insights into their applications in breast cancer research, regenerative medicine, and drug discovery. The collaborative efforts of researchers worldwide are pushing the boundaries of understanding and utilizing 3D cell cultures, opening avenues for transformative therapeutic approaches and personalized medicine. As we delve deeper into the intricacies of these systems, the potential for groundbreaking advancements in medical science becomes increasingly tangible.

## Figures and Tables

**Figure 1 biomedicines-12-00052-f001:**
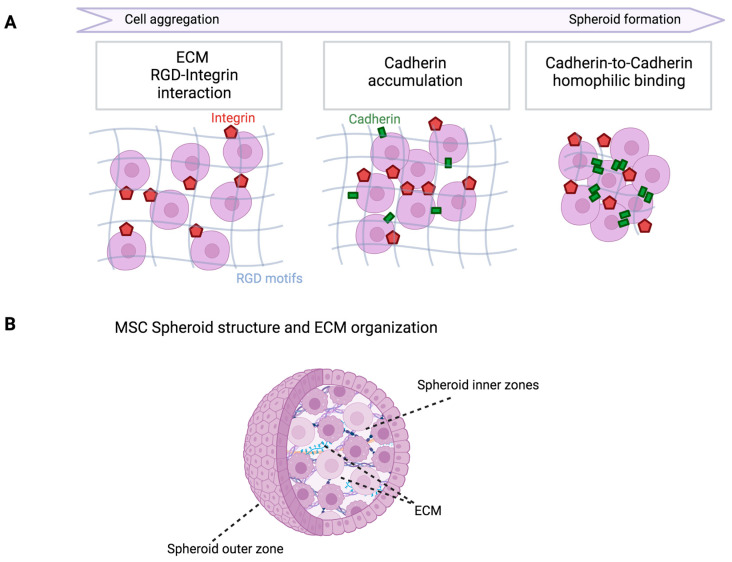
Structure and mechanism of production of mesenchymal stem cell spheroids. (**A**) Three stages of cell aggregation and spheroid development. Cells form loose aggregates at first via the strong interaction of membrane-bound integrin with extracellular matrix arginine–glycine–aspartic (RGD) motifs. Increased cell-to-cell contact causes the cadherin protein to concentrate on the cell membrane and upregulate cadherin expression levels. Dense cell spheroids are formed from cell aggregates in the later phase by homophilic cadherin-to-cadherin interaction. (**B**) MSC spheroids may be structurally classified into inner and outer zones according to their size and the amount of nutrients and oxygen they receive in vitro. To obtain greater spheroid functioning in in vivo environments, consideration should always be given to the oxygen, nutrients, and waste gradients inside the spheroids when choosing the best method for generating spheroids in vitro.

**Figure 2 biomedicines-12-00052-f002:**
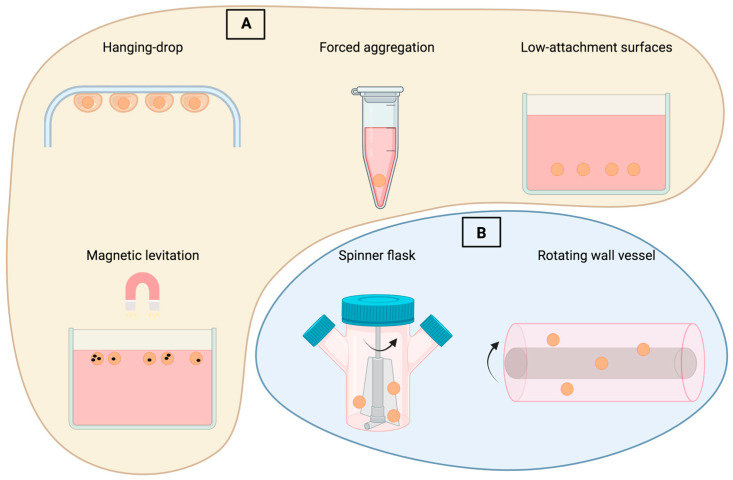
Techniques for producing ex vivo MSC spheroids. (**A**) Static culture techniques: forced aggregation using gravity forces and hanging drop technique. Low-attachment surfaces let cells self-organize into three dimensions. Encouraging intercellular communication by magnetic levitation. (**B**) Dynamic culture: Spinner flask bioreactor system. The rotating wall vessel technique simulates microgravity by constant rotation.

**Table 1 biomedicines-12-00052-t001:** MSC functionalization in 3D spheroids: benefits and drawbacks.

Advantages of 3D MSC Spheroids	Disadvantages of Employing 3D Spheroids
Molecular and immunophenotypic characteristics of MSCs are enhanced	Size variability based on spheroid generation technique
Improved stemness and differentiation potential	Variabilities in the nutrients, waste, and oxygen concentration depending on spheroid size
Improved in vivo survival and homing after infusion	The possibility of necrotic spheroid core development
Improved secretion of anti-inflammatory, anti-fibrotic, anti-apoptotic, mitogenic, and angiogenic factors	The necessity of easy, repeatable, and affordable methods for the large-scale manufacturing of MSC spheroids to facilitate therapeutic needs

## Data Availability

Not applicable.

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
