# Peer review of "Exploring the Three-Dimensional Frontier: Advancements in MSC Spheroids and Their Implications for Breast Cancer and Personalized Regenerative Therapies"

_biomedicines, 2023, doi:10.3390/biomedicines12010052_

Round 1

Reviewer 1 Report

Comments and Suggestions for Authors

Overall synopsis

The title suggests that this review will delve into the development and use of 3D models of MSC cultures and their effect on cancer research broadly. Having read through the manuscript, I can only say that this title is not clearly highlighted and does not form a logical thread throughout this review. The review reads almost as a collection as short synopses of a few papers as well as the author's own unpublished and absent data. This is a shame because this topic could make for a very interesting review with insights on where we currently are and how the future of this sub field may look. Moreover, it was hard to decipher what the authors mean tin many sentences while others were just blatantly incorrect.

I have included some specific comments below but this review requires aa careful rethink and extensive rewriting to provide value to the audience. 

Specific comments

1. Table 1 is missing a heading/legend and clearer column headings

2. Paragraph starting at line 92 is very jumpy. Improve the narrative and flow of this paragraph.

3. Some subheadings might be useful to guide readers. It is currently a monolith of text.

4. Try to avoid adjectives like significant, notably and remarkable, these are over the top.

5. Sentence in line 238 is incomplete.

6. line 259, cancer cannot be heavily malignant. Please refrain from floral language.

7. Line 248. What is meant by uneven expression? Also, this whole sentence is non-sensical. Rethink and rephrase.

8. 252-253. You cannot refer to your own research here without showing the data or referring to a published paper. Also, MCF7 cells are actually quite motile in 2D so specify what you mean. Lastly, the section between 252 and 254 adds nothing to the review and should be left out.

9.267-277. Sentence makes zero sense. You compare MSC monocultures and then measure in MDA?? Incomprehensible, please rethink and rephrase.

10. 269-270 seems like a repeat of the previous sentence in different wording.

11. 259-284 This is a very frustrating paragraph. Firstly, this whole section refers to only 1 paper while TGFb, uPA and PAI are well researched molecules as are co-culture models looking at these molecules and their effects. Secondly, after reading it twice I can still not make head or tails from what is actually being posited here. Thirdly, impossible statements as highlighted in comment 9 and 10 are included. This type of writing extends to the rest of the rest of the review severely limiting the usefulness of this document.

Comments on the Quality of English Language

Spelling and grammar are adequate but language use is problematic with some rather outlandish adjectives being used which are not scientifically sound and for which I have given some examples in my specific comments. 

Author Response

We thank the reviewer for most constructive comments.

All points are now included in the article.

1. headings now present

2. paragraph rewritten

3. subheadings included

4.-8. corrected, sections removed

9.-11. the whole is paragraph rewritten

Reviewer 2 Report

Comments and Suggestions for Authors

A review by Smolinska et al focuses on the benefits and potential applications of MSC spheroids, particularly in the context of breast cancer and customized regenerative therapies.

Specific comments:

1. MSC abbreviation must be explained in abstract.

2. The legend for table 1 is missing. In addition, the concept of this table is unclear - it is just a number of sentences that can be put in the text without the necessity to create a table. 

3. Certain parts of the review are poorly or unconvincingly written, e.g., Conclusions.

Comments on the Quality of English Language

minor revision required

Author Response

We thank the reviewer for the suggestions, now points 1,2 and 3 are corrected in the article. The table includes headers to better explain the meaning of the table and conclusions are rewritten as a whole.

Round 2

Reviewer 1 Report

Comments and Suggestions for Authors

No further comments. All issues have been addressed.